# Patch Antenna-in-Package for 5G Communications with Dual Polarization and High Isolation

**Hugo Santos [1], Pedro Pinho [2],***  **and Henrique Salgado [1]**

[1]   Instituto de Engenharia de Sistemas e Computadores, Tecnologia e Ciência, Faculdade de Engenharia da Universidade do Porto, 4200-465 Porto, Portugal; hmsantos@fe.up.pt (H.S.); hsalgado@fe.up.pt (H.S.)

[2]   Instituto de Telecomunicações and ADEETC, Instituto Superior de Engenharia de Lisboa, 1959-007 Lisboa, Portugal

*    Correspondence: ptpinho@av.it.pt

**Abstract:** In this paper, we describe the design of a dual polarized packaged patch antenna for 5G communications with improved isolation and bandwidth for K-band. We introduce a differential feeding technique and a heuristic-based design of a matching network applied to a single layer patch antenna with parasitic elements. This approach resulted in broader bandwidth, reduced layer count, improved isolation and radiation pattern stability. The results were validated through finite element method (FEM) and method of moments (MoM) simulations. A peak gain of 5 dBi, isolation above 40 dB and a radiation efficiency of 60% were obtained.

**Keywords:** 5G; patch antenna; dual-polarization; high isolation

## 1. Introduction

The European 5G communication band in the frequency range from 24.25 GHz to 27.5 GHz is considered the most adequate solution for limited coverage and high throughput scenarios [1]. Its usage drives the need for new antenna technologies capable of beamforming, polarization diversity and large bandwidths.

Patch antennas are a possible solution for the challenge of 5G K-band antennas, given their size which facilitates package integration, simplicity and ability to support two polarizations in two degenerate modes. Nonetheless, their bandwidth is usually very limited and bandwidth improvement techniques are generally required in broadband communication systems [2].

The most common solution found in the state-of-the-art to overcome bandwidth limitations is the usage of a stacked patch topology. However, such a technique can lead to higher order modes which will degrade radiation pattern stability throughout the band of operation [3]. Furthermore, these higher order modes also compromise isolation in dually polarized antennas. In some cases, a different patch topology has been used to improve isolation between polarizations.

The authors of [4] presented an antenna-in-package solution based on LTCC for 28 GHz. A stacked patch topology was used with a stripline-based feeding network that allowed a wide impedance bandwidth of approximately 3.5 GHz to be obtained. Nonetheless, their approach focused on a single polarized patch antenna which does not allow for polarization diversity usage. Lu et al. presented a patch antenna for 28 GHz based on a parasitic patch antenna on an unbalanced substrate. This approach is based on flip-chip technology and allowed a large bandwidth of approximately 4 GHz to be obtained [5]. Once again, this design was only suitable for single polarization. In [6], the authors proposed a flip-chip structure for a parasitic patch antenna. They used single ended probe feeds for each polarization and were able to achieve a wide bandwidth of approximately 4.5 GHz.

However, as only single-ended feeding was used, the isolation has its worst value of 20 dB at the centre frequency.

In this work, we propose the usage of a patch antenna with parasitic elements in the same layer to improve its bandwidth, reducing the layer count from the typical stacked patches as reported in the previous paragraph. Based on [7], we utilized a differential feeding technique. In this approach, each mode is excited with two feeds with opposing phases, which cancels out higher order modes, reduces feed radiation and reinforces the fundamental degenerate modes $TM_{010}$ and $TM_{100}$ of the patch antenna. Ultimately this improves isolation between polarizations and radiation pattern stability.

## 2. Antenna Design

### 2.1. Patch Antennas

Considering the equivalent transmission line model of a patch antenna, which considers it as an half-wave microstrip resonator, shown in Figure 1, one can calculate the antenna dimensions for a particular design frequency, given the substrate height $h$ and dielectric constant $\epsilon_r$. The patch is seen as a microstrip line whose width should be given as [2]

$$W = \frac{1}{2f_0\sqrt{\mu_0\epsilon_0}}\sqrt{\frac{2}{\epsilon_r+1}} \tag{1}$$

where $f_0$ denotes the desired resonant frequency. The parameters $\mu_0$ and $\epsilon_0$ denote the magnetic permeability and electrical permittivity of vacuum. As this microstrip line is wide enough for radiation to occur and the substrate is thin enough for substrate modes and surface waves to be absent [2], one can calculate its effective dielectric constant as

$$\epsilon_{reff} = \frac{\epsilon_r+1}{2} + \frac{\epsilon_r-1}{2}\left[1+12\frac{h}{W}\right]^{-1/2} \tag{2}$$

Given the effective dielectric constant of the microstrip line, the fringing field extension $\Delta L$ can be given as

$$\Delta L = 0.412 \cdot h \frac{(\epsilon_{reff}+0.3)(\frac{W}{h}+0.264)}{(\epsilon_{reff}-0.258)(\frac{W}{h}+0.8)} \tag{3}$$

This allows us to calculate the actual patch length as

$$L = \frac{1}{2f_0\sqrt{\epsilon_{reff}}\sqrt{\mu_0\epsilon_0}} - 2\Delta L \tag{4}$$

**Figure 1.** Fundamental mode half-wave microstrip resonator model of patch antenna.

This type of antennas can be fed in a variety of ways [2]. In the case of microstrip and probe feeds, the feeding method is direct. However, due to the microstrip discontinuity and probe length, a stray inductance develops which can be hard to compensate in probe feed, mostly in thicker

substrates. Slot feed can be seen as the coupling of two resonant systems, the slot and the patch antenna itself. This feeding method allows more degrees of freedom for tuning and is able to provide wider bandwidths. However, if the slot size becomes electrically large, backside radiation will start developing which distorts the patch radiation pattern and degrades the front-to-back ratio. Proximity coupling feed is also an indirect feeding method as slot feed. It can also be viewed as the coupling of two resonant structures but instead of inductive coupling, they couple through capacitive means. Both these methods are able to provide wider impedance bandwidths but the final end result is the need for higher layer counts which increases fabrication costs.

### 2.2. Rotationally Symmetric Patch with Parasitics

The most common solution found in the state-of-the-art to overcome the bandwidth limitations of the patch antenna is the usage of a stacked patch topology [4,5], as seen in Figure 2. However, such technique usually leads to higher order modes which will degrade radiation pattern stability throughout the band of operation [3]. Furthermore these higher order modes also compromise isolation in dual-polarized antennas. In [6], a different patch topology has been used to improve isolation between polarizations.

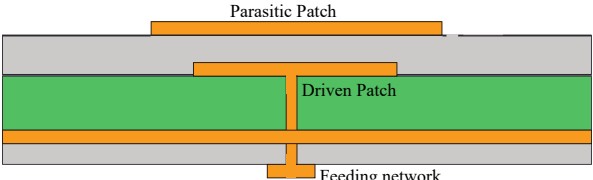

**Figure 2.** Example of parasitic patch antenna topology.

To reduce the layer count and the manufacturing costs a probe-fed patch with parasitic elements in the same layer to improve its bandwidth, as reported in [8], was used. Furthermore, based on [7,9], a differential feeding technique was utilized. As stated before, this approach allows $TM_{010}$ and $TM_{100}$ modes of the patch antenna to be reinforced and have a very low coupling. Such leads to high isolation between orthogonal polarizations and improved radiation pattern stability.

The antenna stackup for land-side die attach packaging is shown in Figure 3, where a Rogers RO4350B ($\epsilon_r = 3.48$) core was used with Rogers RO4450T prepreg ($\epsilon_r = 3.35$). These substrates offer a good compromise between cost, moderate permittivity values and low losses (tan $\delta \approx 0.004$). The metal layers are all composed of 17.5 µm thick copper. Since the patch has to be rotationally symmetric for both polarizations to have the same resonant frequency, we have $W = L$. As the purpose is to design an antenna with dual polarization it is required that $W = L$, which results in a square patch with the two orthogonal modes $TM_{010}$ and $TM_{100}$ having the same frequency, that is they are degenerate. However, it also results in an equation with no analytical solutions for the resonant frequency of the antenna. Therefore a numerical approach was used to determine the initial patch size. The remaining calculation variables for obtaining the resonant frequency as a function of $W = L$ are $h = 0.762$ mm and $\epsilon_r = 3.48$, since the patch is designed in $M_1$ layer, being $M_2$ the ground plane. To obtain the length and width of the rotationally symmetric patch $L = W$ a simple patch length was first calculated using the method described in Section 2.1. The result of such calculation can then be used as the initial guess of a Newton algorithm approach as described in [10]. The Newton algorithm in this case was set to obtain the root of the objective function

$$f(L) = f_0 - \frac{1}{2(L + 2\Delta L)\sqrt{\epsilon_{reff}}\sqrt{\mu_0\epsilon_0}} \tag{5}$$

where $f_0 = 26$ GHz is the desired resonant frequency. The values for $\epsilon_{reff}$ and $\Delta L$ can be obtained during the algorithm from Equations (2) and (3), respectively, by making $W = L$. Using this algorithm a resonant frequency of $f_0 = 26$ GHz was obtained for a patch size of $W = L \approx 2.75$ mm.

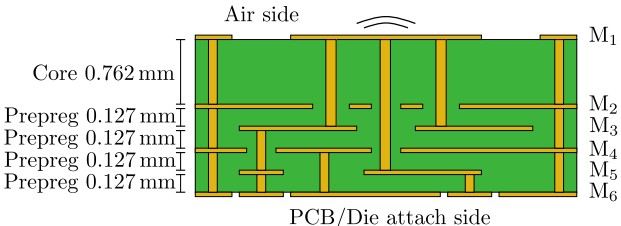

**Figure 3.** Antenna stackup.

The patch and its parasitics for bandwidth enhancement are designed in $M_1$ metal layer, as shown in Figure 4. The vertically polarized stream (port 1) is fed from $M_3$ and the horizontal (port 2) from $M_5$ metal layer. In each feeding network a lossless half-wavelength $50\,\Omega$ transmission line was connected between the two corresponding balanced ports to obtain the needed $180\,°$ phase shift for differential feeding. Another $50\,\Omega$ lossless line was connected to one of the balanced ports in each polarization, to evaluate the single-ended reflection coefficient. Considering the previously calculated patch size, it was set to $ps = 2.75\,\text{mm}$. As stated in [8], optimization is needed for determining the size of the parasitics and their distance to the driven patch, which should be kept $pg < 2.5$ h. The design of [8] was scaled down for our patch size and used as an initial guess of a gradient based optimizer combined with the Method of Moments simulation in ADS. The objective was to maximize the bandwidth centred at 26 GHz in both ports. After optimization, the values of $ps = 2.75\,\text{mm}$ (patch size), $fo = 0.8\,\text{mm}$ (probe feed offset from patch centre), $pl = 1.7\,\text{mm}$ (parasitic length), $pw = 1.2\,\text{mm}$ (parasitic width), $pg = 0.1\,\text{mm}$ (parasitic gap) resulted in the plot of Figure 5, where a low reflection coefficient at the centre frequency of 26 GHz, was obtained at both ports. However, a limited bandwidth of approximately 2.5 GHz can be verified, which falls short from the needed 3.25 GHz.

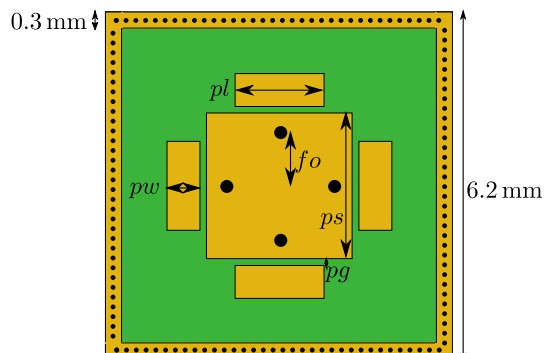

**Figure 4.** Antenna layout.

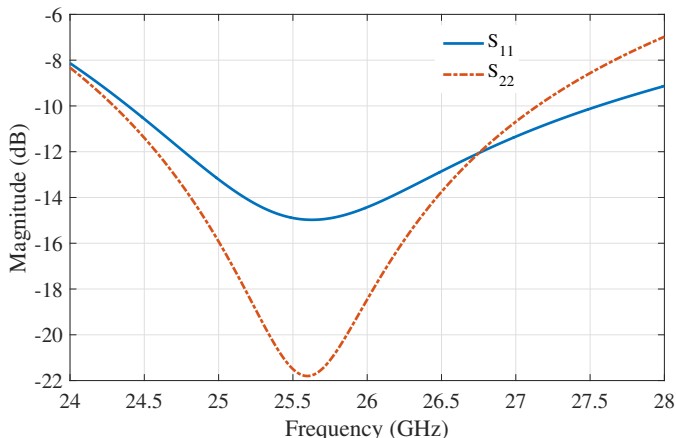

**Figure 5.** $S_{11}$ and $S_{22}$ magnitudes as a function of frequency for the patch antenna without matching network.

To overcome the bandwidth limitation a matching network was designed. Since the transition from the antenna feeding networks to the PCB presents some parasitics, the matching network must be designed after such transition is modelled.

## 3. PCB-to-Package Transition

The transition between the hosting PCB and the antenna-in-package is designed as shown in the 3D model of Figure 6. The transition is simply a connection between a microstrip on the host PCB and a stripline with via stitching on $M_5$ layer of the package. To connect to the feeding network on $M_3$ an internal transition between $M_5$ and $M_3$ was required to be designed as well. Port 1 is placed in the microstrip on the PCB and port 2 in the stripline inside the package.

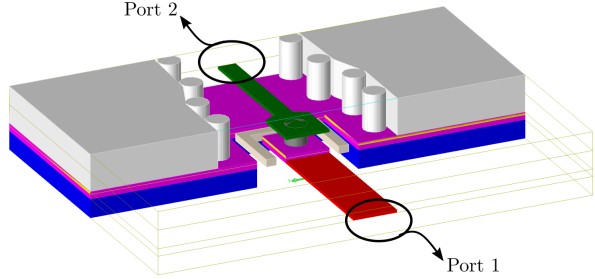

**Figure 6.** PCB to package transition.

According to [11], the microstrip dimensions for a certain impedance $Z_0$ can be obtained by calculating two factors $A$ and $B$ as

$$A = \frac{Z_0}{60}\sqrt{\frac{\epsilon_r + 1}{2}} + \frac{\epsilon_r - 1}{\epsilon_r + 1}\left(0.23 + \frac{0.11}{\epsilon_r}\right) \tag{6}$$

and

$$B = \frac{377\pi}{2Z_0\sqrt{\epsilon_r}} \tag{7}$$

which for $W/d > 2$ we have

$$W/d = \frac{8e^A}{e^{2A} - 2} \tag{8}$$

and for $W/d \leq 2$

$$W/d = \frac{2}{\pi}\left[B - 1 - \ln(2B - 1) + \frac{\epsilon_r - 1}{2\epsilon_r}\left\{\ln(B - 1) + 0.39 - \frac{0.61}{\epsilon_r}\right\}\right] \tag{9}$$

where $W$ is the microstrip width and $d$ the substrate height. To obtain the stripline width, according to [11], one can write

$$\frac{W}{b} = \begin{cases} x & \text{for } \sqrt{\epsilon_r}Z_0 < 120 \\ 0.85 - \sqrt{0.6 - x} & \text{for } \sqrt{\epsilon_r}Z_0 < 120 \end{cases} \tag{10}$$

where $b$ is the substrate height, $Z_0$ the desired characteristic impedance and

$$x = \frac{30\pi}{\sqrt{\epsilon_r}Z_0} - 0.441 \tag{11}$$

These formulas for stripline assume that the transmission line is of zero thickness and that it sits in the exact centre of the two ground planes. The latter assumption can be considered as acceptable given the stackup of Figure 3. However, since the stripline in the real case has no zero

thickness, its impedance was optimized in ADS Controlled Impedance Line Designer as shown in the cross-section of the stripline structure of Figure 7.

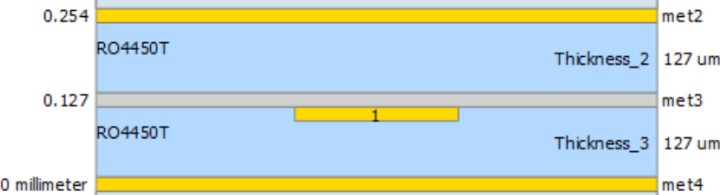

**Figure 7.** ADS controlled impedance line designer setup for obtaining stripline dimensions.

After FEM simulation in ADS, the S-parameters plotted in Figures 8 and 9 were obtained where it can be seen that an acceptable return and insertion losses were obtained which validates the technique.

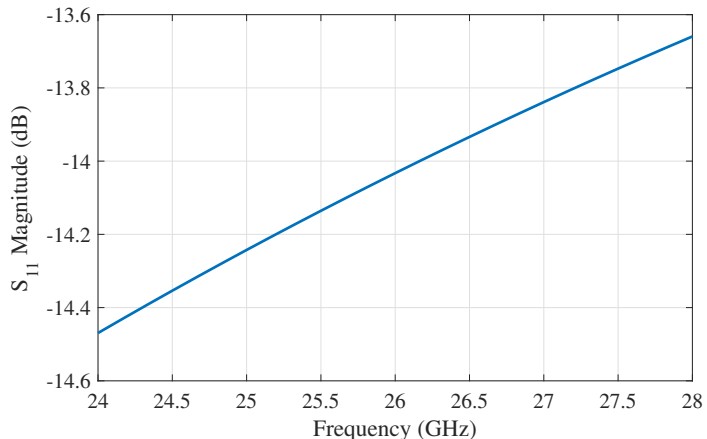

**Figure 8.** $S_{11}$ magnitude as a function of frequency for the microstrip input of the PCB (port 1) to package stripline transition (port 2).

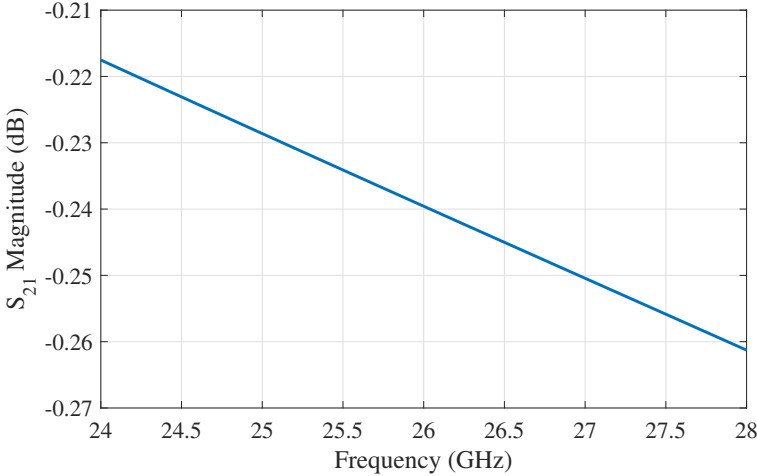

**Figure 9.** $S_{21}$ magnitude as a function of frequency for the microstrip input of the PCB (port 1) to package stripline transition (port 2).

## 4. Final Assembly

Due to the reduced bandwidth obtained in the modelling described in Section 2, the patch antenna is re-optimized to achieve peak gain instead of maximizing impedance bandwidth, as matching networks will later be connected for each polarization. The resulting values for peak gain obtained after the gradient based optimizer are $ps = 2.83$ mm, $fo = 0.8$ mm, $pl = 1.7$ mm, $pw = 0.7$ mm,

$pg = 0.1$ mm. Matching networks are later introduced to obtain $50\,\Omega$ input impedance at the package input.

In [12] a method with a dual-frequency impedance transformer, composed of two arbitrary cascaded transmission lines, was reported to match complex impedances in a wide bandwidth. The proposed matching network circuit for the antenna is shown in Figure 10, where two cascaded transmission lines and two reactance cancellation stubs can be seen. The values of the electrical parameters for both matching networks are obtained by resorting to a particle swarm optimizer (PSO) algorithm, because the method reported in [12] does not consider load impedance variation with frequency, therefore rendering analytical calculations useless for our case. The upper bound for the impedance of each transmission line is obtained by calculating the characteristic impedance of a stripline with the minimum trace width of $100\,\mu$m, for easier fabrication. This impedance was calculated using ADS Controlled Impedance Line Designer and the value obtained was $Z_{max} = 53.8\,\Omega$. The remaining PSO parameters are defined as recommended in [13], where the number of particles $N = 100$, the social and cognitive coefficients $c_1 = c_2 = 2$ and the inertia coefficient $w = 1$. The objective function is defined so that its value reduces as more simulation points for the magnitude of the $S_{11}$ are below the $-10$ dB. Mathematically this can be described as summing all the points that lie above the $-10$ dB line and therefore are unmatched. The objective function can be normalized by dividing the sum of points above $-10$ dB and dividing by the total simulated points $k$. In mathematical terms we can write

$$OF = \frac{1}{k}\sum_{i=1}^{k} a_i \tag{12}$$

where $i$ denotes the index of the simulation point corresponding to a simulation frequency within the considered span of 24.25 to 27.5 GHz and $a_i$ is given as

$$a_i = \begin{cases} 0 & \text{for } |S_{11i}|_{\text{dB}} < -10 \\ 1 & \text{for } |S_{11i}|_{\text{dB}} \geq -10 \end{cases} \tag{13}$$

in which $S_{11i}$ denotes the simulated reflection coefficient at frequency index $i$. After optimization the values for the transmission line electrical parameters are obtained as given in Table 1 for both the V-Pol and H-Pol inputs of the antenna. Despite the rotational symmetry of the antenna, the matching networks have to be different because of an added transition between $M_3$ and $M_5$ metal layers, which results in an asymmetric S-parameter matrix. This transition serves as a connection between $M_3$ layer and the package to PCB transition.

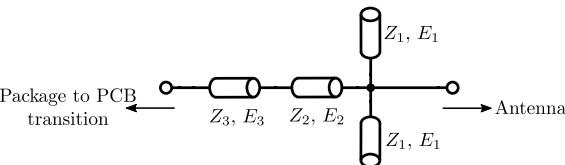

**Figure 10.** Matching network schematic.

**Table 1.** Matching network transmission line electrical parameters at 26 GHz.

| Polarization | $Z_1$ | $E_1$ | $Z_2$ | $E_2$ | $Z_3$ | $E_3$ |
|---|---|---|---|---|---|---|
| V-Pol | $53.8\,\Omega$ | $50\,^\circ$ | $53.8\,\Omega$ | $0\,^\circ$ | $20.1\,\Omega$ | $107\,^\circ$ |
| H-Pol | $53.8\,\Omega$ | $58\,^\circ$ | $53.8\,\Omega$ | $26.8\,^\circ$ | $16\,\Omega$ | $43.4\,^\circ$ |

The final layout is presented in Figure 11, where the values of Table 1 were mapped to striplines by resorting to Equation (10) and ADS Controlled Impedance Line Designer. In this layout, the transition between metal layers $M_3$ and $M_5$ is also shown, as well as the package to PCB transition. A 3D representation of the model is shown in Figure 12.

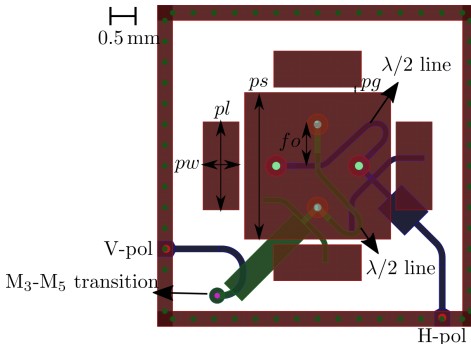

**Figure 11.** Final layout with feeding and matching networks, inter layer and PCB to package transitions (ground planes and grounding vias omitted).

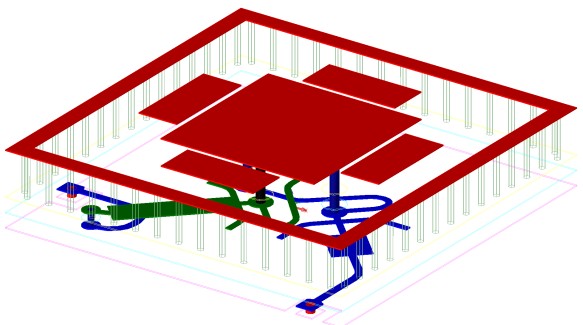

**Figure 12.** 3D model of full layout (ground planes and grounding vias omitted).

The full system is simulated using ADS Momentum and the magnitude of the S-parameters on V-pol (port 1) and H-pol (port 2) ports when fed from a $50\,\Omega$ microstrip line on the host PCB, are shown in Figure 13, where it can be seen that not only the reflection coefficient magnitude is below $-10\,\text{dB}$ in both ports, but also the isolation is greater than $40\,\text{dB}$ over the bandwidth of $24.25\,\text{GHz}$ to $27.5\,\text{GHz}$. The E-Plane and H-Plane radiation patterns at $24.25\,\text{GHz}$, $26\,\text{GHz}$ and $27\,\text{GHz}$ are shown in Figures 14–16, respectively. It can be seen that the radiation pattern suffers practically no change from the beginning to the end of the 5G K-band. Such is due to the differential feed topology which reinforces the fundamental $TM_{100}$ and $TM_{010}$ degenerate modes and cancels higher order modes resulting in a frequency stable radiation pattern. A radiation efficiency of approximately $60\,\%$ and a peak gain of $5\,\text{dBi}$ were observed, in simulation, at both V-pol and H-pol ports.

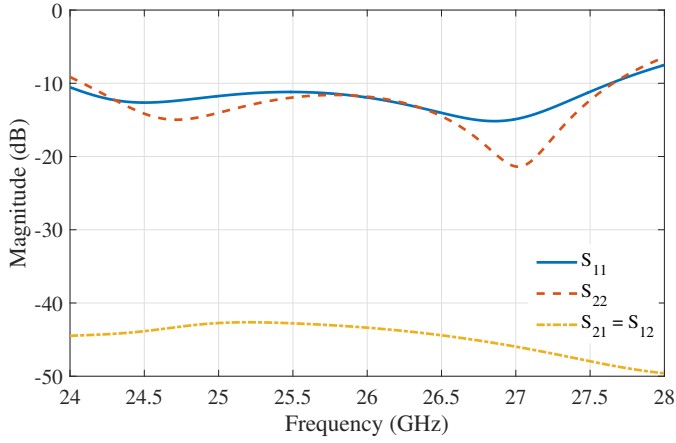

**Figure 13.** Simulated results for the magnitude of V-Pol (port 1) and H-Pol (port 2) S-parameters.

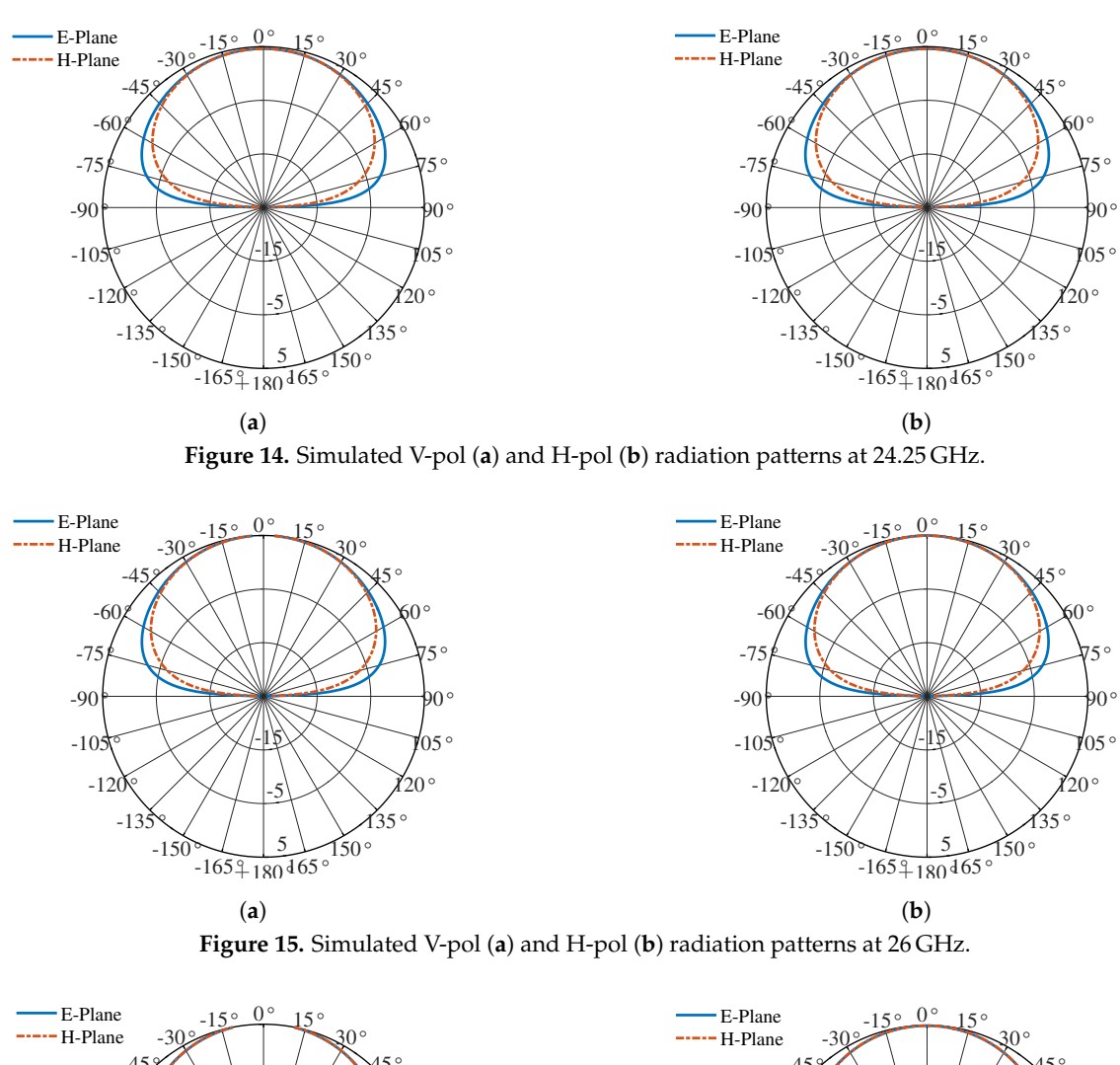

**Figure 14.** Simulated V-pol (**a**) and H-pol (**b**) radiation patterns at 24.25 GHz.

**Figure 15.** Simulated V-pol (**a**) and H-pol (**b**) radiation patterns at 26 GHz.

**Figure 16.** Simulated V-pol (**a**) and H-pol (**b**) radiation patterns at 27.5 GHz.

## 5. Conclusions

In this work, a new topology of patch antenna-in-package was shown. The usage of differential feeding was shown to be capable of achieving isolation in the order of 40 dB, by exploiting the cancellation of common mode interference between adjacent probe feeds. Furthermore it was shown that such technique results in a frequency stable radiation pattern. An heuristic approach was used for the design of a two section matching network that allowed the impedance bandwidth improvement of the antenna cell.

Nonetheless, despite reducing the number of layers for the antenna by using coplanar parasitic elements instead of a parasitic patch antenna on top, extra complexity was added in the matching

process. This added complexity comes with higher fabrication costs that can ultimately cancel the cost reduction of a single layer patch antenna. For this reason, cheaper packaging solutions can be exploited with this topology to significantly reduce fabrication costs.

**Author Contributions:** Conceptualization, H.S. (Hugo Santos); methodology, H.S. (Hugo Santos); software, H.S. (Hugo Santos); validation, H.S. (Hugo Santos), P.P. and H.S. (Henrique Salgado); formal analysis, H.S. (Henrique Salgado); investigation, H.S. (Henrique Salgado); resources, H.S. (Henrique Salgado); data curation, H.S. (Hugo Santos); writing—original draft preparation, H.S. (Hugo Santos); writing—review and editing, P.P. and H.S. (Henrique Salgado); visualization, H.S. (Hugo Santos); supervision, H.S. (Henrique Salgado); project administration, H.S. (Henrique Salgado); funding acquisition, H.S. (Henrique Salgado). All authors have read and agreed to the published version of the manuscript.

**Funding:** This work is funded by FCT/MCTES through national funds and when applicable co-funded EU funds under the project UIDB/50008/2020-UIDP/50008/2020.

**Conflicts of Interest:** The authors declare no conflict of interest.

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
