# Peer review of "Patch Antenna-in-Package for 5G Communications with Dual Polarization and High Isolation"

_electronics, doi:10.3390/electronics9081223_

Round 1
Reviewer 1 Report
Paper can be accepted after the following corrections:
- All parameters appearing in equations should be carefully defined.
- Figure 8 is not clear. Please re-draw.
- Conclusions should be developed to clearly present quantitatively the most important achievement of the research. Please also indicate the further research possibilities.
Author Response
Reviewer#1,
Concern # 1: All parameters appearing in equations should be carefully defined.
Author action: Reviewed missing parameter definitions. Added
Concern # 2: Figure 8 is not clear. Please re-draw.
Author action: Figure re-drawn and explanatory text added.
Concern # 3: Conclusions should be developed to clearly present quantitatively the most important achievement of the research. Please also indicate the further research possibilities
Author response: The authors thank the reviewer for the recommendation.
Author action: Extended conclusions and included future work possibilities.

Reviewer 2 Report
This paper presents an improved patch antenna for the 5G. The description of the proposed design is well organized. Meanwhile, the following drawbacks need to be considered in the revision. First of all, it is currently hard to evaluate the novelty of the proposed design, as it is not compared with the preceding ones in the literature. In addition, Section 1 does not well review the recent works. In short, the paper is only describing and analyzing its own design, while it is not taking into account the contemporary techniques sufficiently.
Author Response
Concern # 1: This paper presents an improved patch antenna for the 5G. The description of the proposed design is well organized.
Author response: Thank you for your opinion.
Concern # 2: First of all, it is currently hard to evaluate the novelty of the proposed design, as it is not compared with the preceding ones in the literature. In addition, Section 1 does not well review the recent works.
Author response: The authors agree with such revision.
Author action: Added new paragraph detailing the analysis of other state-of-the-art works.
Concern # 3: In short, the paper is only describing and analyzing its own design, while it is not taking into account the contemporary techniques sufficiently.
Author response: The authors agree with such revision.
Author action: Added new paragraph detailing the analysis of other state-of-the-art works (same as above)

Round 2
Reviewer 2 Report
The concerns of the reviewer have been addressed well.